# Development of Soft Rice Lines by Regulating Amylose Content via Editing the 5′UTR of the *Wx* Gene

**DOI:** 10.3390/ijms231810517

**Published:** 2022-09-10

**Authors:** Jinlian Yang, Xinying Guo, Xuan Wang, Yaoyu Fang, Fang Liu, Baoxiang Qin, Rongbai Li

**Affiliations:** State Key Laboratory for Conservation and Utilization of Subtropical Agro-Bioresources, College of Agriculture, Guangxi University, Nanning 530004, China

**Keywords:** rice, CRISPR/Cas9, *Wx* gene, eating and cooking quality (ECQ), amylose content

## Abstract

The type of soft rice with low amylose content (AC) is more and more favored by consumers for its better eating and cooking quality, as people’s quality of life continuously improves in China. The *Wx* gene regulates the AC of rice grains, thus affecting the degree of softness of the rice. Mei Meng B (MMB), Tian Kang B (TKB), and DR462 are three indica rice maintained lines with good morphological characters, but also with undesirably high AC. Therefore, CRISPR/Cas9 technology was used to edit the *Wx* gene of these lines to create a batch of soft rice breeding materials. New gene-edited lines MMB-10-2, TKB-21-12, and DR462-9-9, derived from the above parental lines, respectively, were selected in the T_2_ generations, with an AC of 17.2%, 16.8%, and 17.8%, and gel consistency (GC) of 78.6 mm, 77.4 mm, and 79.6 mm, respectively. The rapid viscosity analysis (RVA) spectrum showed that the three edited lines had a better eating quality as compared to the corresponding wild type, and showing new characteristics, different from the high-quality soft rice popular in the market. There was no significant difference in the main agronomic traits in the three edited lines compared to the corresponding wild types. Moreover, the chalkiness of DR462-9-9 was reduced, resulting in an improved appearance of its polished rice. The present study created soft rice germplasms for breeding improved quality hybrid rice, without changing the excellent traits of their corresponding wild type varieties.

## 1. Introduction

Along with the improvement of our national living standards, consumers have higher requirements for the eating and cooking quality (ECQ) of rice. Rice with low AC has the advantages of palatability, good puffing properties, softness, elasticity, a low degree of retrogradation and hardening after cooling, and does not rot easily [1]. As the main component of the rice endosperm, AC is an important index for evaluating the ECQ of rice, which determines the taste and cooking properties of rice [2,3]. Using their AC, rice varieties can be divided into five categories: high (25~33%), medium (20~25%), low (12~20%), very low (5~12%), and waxy (0~5%) [4]. Countries and regions have different cultural favor for rice categories; therefore, it is necessary to develop multiple types of rice.

Since the 1980s, Japan has taken the lead in cultivating rice varieties with low amylose content [5]. In recent years, attention was gradually given to the development of cultivated rice varieties with low AC in rice breeding research in China, and a number of studies on the genetics, agronomy, processing quality and other aspects have been carried out in this regard [6,7,8]. A series of soft rice japonica varieties such as Yunjing 20, Yunjing 29, Yunjing 37, and Yunjing 41 bred in Yunnan; Huruan 1212 and songxiangjing1018 bred in Shanghai; Nanjing 46, Nanjing 9108, Fengjing 1606, and Wuxiangjing 113 bred in Jiangsu; and Jia 58 bred in Zhejiang have been popularized [9,10,11]. However, at present, the germplasm resources of rice with low AC that can be used as restorers and maintainers are still limited.

The waxy gene (*Wx*) encodes granule-bound starch synthase I (GBSSI), which is the key gene regulating amylose synthesis. The *Wx* gene directly affects AC in rice endosperm, thus affecting the degree of softness of rice. The CRISPR/Cas9 gene editing system has the advantages of high efficiency and easy operation and has been widely used in crop genetic improvement and breeding [12]. Previous studies have shown that the CRISPR/Cas9 system was used in the research on the *Wx* allele in rice and a series of germplasms were created for breeding [8,13,14,15]. Most researches focused on the editing of coding sequences to eliminate *Wx* expression, in order to generate glutinous rice. Teng (2021) used the CRISPR/Cas9 system to mediate the editing of the *Wx* gene in a Photothermosensitive Genic-Male-Sterile (PTGMS) line Y58S, which caused ultra-low AC mutations that produced a PTGMS glutinous rice strain with excellent waxiness [16]. Fu (2022) developed a rapid and highly efficient strategy through the CRISPR/Cas9 gene-editing system for generating *Wx* mutants from the background of five different rice varieties, which significantly reduced the AC and starch viscosity but did not affect the major agronomic traits [17]. Liu (2022) performed the targeted deletion of the first intron of the *Wx*^b^ allele via CRISPR/Cas9. The grain AC of mutant lines significantly increased from 13.0% to 24.0% [15].

In addition to creating glutinous rice, Huang (2020) generated novel Wx alleles by ed-iting the region of the Wxb promoter, in turn fine-tuning of Wx expression [3]. In this study, Meimeng B (MMB), Tiankang B (TKB), and DR462, which are high-quality indica rice parent materials commonly used in breeding, but with a hard rice quality, were used as test materials. They have high a yield performance but also high AC and low GC, resulting in poorer rice quality. It is of great significance to improve the quality of rice and increase the value of breeding and utilization by appropriately changing the AC and GC contents of these lines. However, the direct editing of the coding region of the *Wx* gene by using CRISPR/Cas9 often results in too low an AC content in edited lines, which directly results in glutinous rice lines. This does not meet the AC content requirement for soft rice. Therefore, in this study, by using CRISPR/Cas9 to edit the intron splice site (5′UISS) in the 5′UTR region of the *Wx* gene, the expression of the *Wx* gene was appropriately reduced, but the original protein coding sequence of the *Wx* gene was not changed. Thus, rice gene editing lines with lower AC and GC can be obtained, which can provide high-quality resources for the improvement of new rice varieties.

## 2. Results

### 2.1. Construction of CRISPR/Cas9-Wx Vector

According to the sequence of the parent *Wx* gene (*LOC_Os06g04200*), the CRISPR/Cas9 vector was constructed by targeting the intron splicing site (5′UISS) in the 5′UTR region of *Wx* (Figure 1A). As confirmed by sequencing and alignment, the sequences of target sites in MMB, TKB, and DR462 were consistent with the designed sequences (Figure 1B), indicating that MMB, TKB, and DR462 were suitable for editing.

A U6a-sgRNA expression cassette of the *Wx* gene was obtained through overlapping PCR, and the amplification product of the target construction fragment was 629 bp in length (Figure 2A). The constructed Cas9/sgRNA expression vector was transformed, cultured, and a single colony was selected. The primer PB-L/PB-R were used for PCR amplification and sequencing. The gel electrophoresis results showed that the *Wx* gene Cas9/sgRNA expression vector was present in 16 out of 20 colonies (Figure 2B), indicating that the CRISPR/Cas9-*Wx* expression vector was successfully constructed.

### 2.2. Genetic Transformation of MMB, TKB, and DR462

The vector was introduced into MMB, TKB, and DR462 rice through *agrobacterium*-mediated transformation. Calli were induced from mature embryos of rice seeds after co-culture (Figure 3A). After two rounds of hygromycin resistance screening (Figure 3B–D), the well-growing calli were selected for plantlet differentiation and rooting (Figure 3E,F). As a result, a total of 30, 33, and 34 plants from MMB, TKB, and DR462, respectively, were obtained, which were regarded as the T_0_ generation.

### 2.3. Mutation Analysis of T_0_ Transformants

T_0_ plants were confirmed with hygromycin resistance marker HPT-F/HPT-R. The amplification results of the target fragments of the T_0_ plants are shown in Figure 4. There were 24, 26, and 26 positive transformed plants from MMB, TKB, and DR462, respectively, with the mutation rate of 80.0%, 78.8%, and 76.5%, respectively (Figure 4).

The genotype analysis of target mutations in the T_0_ generation of MMB, TKB, and DR462 showed that there were four types of mutations, including a homozygous mutant, bi-allelic mutant, heterozygous mutant, and wild-type mutant. Among them, the homozygous mutants accounted for the highest proportion. Respectively, there were 8, 9, and 9 with 33.3%, 34.6%, and 34.6% of the homozygous mutant rate in the T_0_ generation of MMB, TKB, and DR462 (Table 1), indicating that this target has a high editing efficiency and is an ideal target for editing the *Wx* site. The analysis of target mutation types showed that most of the mutations were deletions only, with deletions ranging from 2 to 26 bases. Specifically, mutant MMB-10 from MMB showed a deletion of 11 bases, mutant TKB-21 from TKB showed a deletion of 26 bases, including an intron splice site (GT), while mutant DR462-9 from DR462 showed a deletion of two bases (Figure 5).

### 2.4. Screening of T-DNA-Free Plants in T_1_ Generations

The homozygous mutant plants MMB-10, TKB-21, and DR462-9 of the T_0_ generation were selfed to develop the T_1_ lines for T-DNA-free detection. The result showed that seven out of 20 T_1_ plants from the line MMB-10 did not carry exogenous T-DNA fragments, all of which were homozygous by sequencing. Five out of 20 T_1_ plants from TKB-21 did not carry exogenous T-DNA fragments and three of which were homozygous by sequencing. Four out of 20 T_1_ plants from DR462-9 did not carry exogenous T-DNA fragments, all of which were homozygous by sequencing. Subsequently, these T-DNA-free homozygous mutants from MMB-10, TKB-21, and DR462-9 were grown and managed under greenhouse conditions (22 °C~25 °C) for harvesting the T_2_ seeds of each line.

### 2.5. Quantitative Analysis of Wx Gene Expression in Mutant Lines in the T_2_ Generation

The expression of the *Wx* Gene in partial homozygous mutant lines from the T_2_ generation was detected by qRT-PCR, including five lines from MMB, three lines from TKB, and four lines from DR462. The results showed that the expression of *Wx* gene in mutant lines was significantly decreased as compared with the parents MMB, TKB, and DR462, indicating that the transcription efficiency of the *Wx* gene was inhibited (Figure 6).

### 2.6. Determination of AC in Mutant Lines in the T_2_ Generation

As to the wild types, the ACs in MMB, TKB, and DR462 were 27.8%, 26.0%, and 25.4%, respectively. By comparison, the AC in the homozygous mutant lines decreased significantly (Figure 7). Among them, MMB-10-2, TKB-21-12, and DR462-9-9 had an AC of 17.2%, 16.8%, and 17.8%, reaching the first-grade high-quality standard concerning AC (13~18%) [18]. The results indicated that gene editing could effectively reduce the AC, so as to obtain lines with an ideal AC.

### 2.7. Determination of Gel Consistency (GC) in Mutant Lines in the T_2_ Generation

Homozygous mutant lines MMB-10-2, TKB-21-12, and DR462-9-9 showed a GC of 78.6 mm, 77.4 mm, and 79.6 mm, while their parents MMB, TKB, and DR462 showed a GC of 60.2 mm, 61.2 mm, and 61.4 mm, respectively, in the analysis. The results indicated that the GC in the mutant lines was significantly increased compared with their respective wild type, and far exceeded the first-grade high-quality standard concerning GC (65 mm) [19] (Table 2).

### 2.8. Analysis of RVA Profile in Mutant Lines in the T_2_ Generation

The RVA analysis results of the edited lines showed that, in comparison with the parent MMB, the mutant line MMB-10-2 showed an increased breakdown viscosity (BDV) (994.0 cp from 355.0 cp), decreased setback viscosity (SBV) (1366.3 cp from 2353.6 cp), decreased consistency viscosity (CSV) (2360.3 cp from 2708.6 cp), and decreased viscosity index (Table 3; Figure 8A). In comparison with TKB, TKB-21-12 showed increased BDV(1533.6 cp from 641.3 cp), decreased SBV (729.6 cp from 1978.0 cp), decreased CSV (2263.3 cp from 2619.3 cp), and decreased viscosity index (Table 3; Figure 8B). In comparison with DR462, DR462-9-9 showed increased BDV (1293.3 cp from 508.7 cp), increased CSV(2738.0 cp from 1966.3 cp), and increased viscosity index, while maintaining an equivalent level in BDV (1444.6 cp from 1457.6 cp) (Table 3; Figure 8C).

Previous studies have demonstrated that BDV, SBV, and CSV are closely related to the hardness and viscosity of rice. Rice with higher BDV, lower SBV, and lower CSV appeared to be softer and more elastic [18]. The results indicated that the ECQ of the homozygous mutant lines was obviously improved as compared to the wild types.

### 2.9. Analysis of the Main Agronomic Traits in Mutant Lines in the T_2_ Generation

To evaluate the selected mutant lines, we surveyed six main agronomic traits of MMB-10-2, TKB-21-12, and DR462-9-9 and their respective wild types. The results revealed that there was no significant difference between the mutant lines and their respective wild types, regarding traits including plant height, 1000-grain weight, panicle length, grain number per panicle, grain set rate, and effective panicle number (Table 4; Figure 9A,C,E). The results indicated that the mutations in the *Wx* genes do not change other agronomic traits. In addition, the grain appearance and chalkiness were observed and compared. The grain of MMB-10-2 and TKB-21-12 was not significantly different from the wild type in chalkiness, while they had a slight decrease in transparency and an increase in whiteness. The grain of DR462-9-9 had a decrease in chalkiness and chalky grain rate, and an increase in whiteness (Figure 9B,D,F).

## 3. Discussion

### 3.1. 5′UTR Region Regulates the Expression of Wx Gene

Previous studies have shown that mutation types such as base deletion and insertion in the non-coding region of *Wx* gene will affect its expression level, and then affect AC in the endosperm [20,21]. This study also revealed such a consistent conclusion. When the 5′UISS of the *Wx* gene was edited, the AC in the three elite indica rice lines was downregulated significantly (Figure 6). Different editing modes of the *Wx* gene have different effects on AC. For example, MMB-10 had a deletion of 11 bp, producing the progeny line MMB-10-2 with an AC of 17.4%; TKB-21 had a deletion of 26 bp, including a splice site (GT), creating the progeny line TKB-21-12 with an AC of 16.8%. In addition, mutations near 5′UISS also have a significant effect on the expression of *Wx*. For example, DR462-9 had a deletion of 2 bp near 5′UISS of the *Wx* gene, which led to the change of mRNA conformation and affected the correct splicing of introns, resulting in a decrease of AC, from 25.4% in the parent DR462 to 17.8% in the progeny line DR462-9-9. It could be confirmed that 5′UISS plays an important role in regulating the *Wx* gene.

There are relevant studies on the 5′UTR region of the *Wx* gene. Cai (1997, 2000) found an intron in the 5′UTR region of the *Wx* gene related to regulation in Indica Rice 232 and further found that the 16 base mutations in the first intron of the *Wx* gene can affect the function and efficiency of the its splice site, resulting in a decrease in the expression level of mature mRNA, thus reducing AC [22,23]. Cheng (2001) found that the natural mutation of G→T on the splice site at the 5′ end of the first intron of the *Wx* gene was the main reason for the decline of the *Wx* gene expression level and AC in rice varieties with medium and low AC [24]. In the study of highland barley starch synthesis, Li (2015) found that there are a large number of polymorphic sites in the 5′UTR region of the *Wx* gene, including large fragments of insertion and deletion. A deletion of about 400 bp in the 5′UTR region will reduce gene expression, resulting in the reduction of amylose content, and affecting the physicochemical and functional characteristics of starch [25].

### 3.2. Application of CRISPR/Cas9 System in Wx Gene Editing

Most studies using the CRISPR/Cas9 system to edit the *Wx* gene of rice directly turn the varieties into waxy ones [26]. Fan (2019) reduced the AC of variety Xiushui 134 from 19.78% to a glutinous rice level (AC < 2%) through gene editing [27]. Wang (2019) knocked out the *Wx* gene of rice line 209B and successfully obtained waxy maintainer *Wx* 209B with a lower AC. Then, the waxy male sterile line *Wx* 209A was developed using *Wx* 209B as male parent and 209A as female parent [28]. The above results show that editing the coding region of *Wx* Gene in rice will “kill” the gene, resulting in mutant lines with a very low AC content and glutinous rice with a high viscosity, which does not meet the requirement of consumers for soft rice quality. Therefore, accurately reducing AC and improving GC are the essence of creating germplasm resources for soft rice breeding. Zeng (2020) edited the 5′UISS locus in the *Wx*b 5′UTRs region of japonica rice material and obtained new germplasms with soft rice traits, and with a very low AC of 9.8~11.5% [29]. Similarly, this study also chose 5′UISS as the target in three indica rice lines with high AC content, and obtained homozygous mutant lines with an AC between 16.8% and 17.8%. The AC of the mutant lines obtained in this study is higher than that of Zeng (2020) and more suitable for consumers. The reasons for this may be (1) less base deletion, resulting in a lower splicing efficiency of mRNA; (2) nucleotide polymorphisms exist between *Wx*^a^ in indica rice and *Wx*^b^ in japonica rice, resulting in differences in the stability of gene mRNA splicing efficiency [30]. Even so, the three mutant lines in the T_2_ generation obtained in this study, MMB-10-2, TKB-21-12, and DR462-9-9, had an AC of 17.2%, 16.8%, and 17.8%, reaching the first-grade high-quality standard concerning AC (13~18%).

### 3.3. Effects on Agronomic Traits of Plants When Editing the 5′UTR of Wx Gene

Gene editing may change the promoter sequence of the target gene, thereby affecting the agronomic traits of the plants [31,32]. Wang (2021) chose a target at the 5′UTR region of the *Wx* Gene of japonica rice variety Jiahua 1, and the mutant lines had main agronomic traits significantly different from that of the wild type, including a higher plant height, increased 1000-grain weight, shortened panicle length, and reduced seed setting rate [33]. In this study, the selected mutant lines in the T_2_ generation had no significant differences in their main agronomic traits compared with the wild type and retained the excellent traits of the wild type. The whiteness of the endosperm in mutant lines increased, which may have been due to the changes in grain appearance caused by the decrease of AC and the increase of GC (Figure 9).

### 3.4. Advantages of CRISPR/Cas9 Technology in Improving ECQ of Rice

In traditional breeding, continuous backcross is usually used to introduce the *Wx* gene into rice varieties, to improve the ECQ of rice. However, this is time-consuming and it is difficult to break the linkage with undesirable traits. Subsequently, the RNAi, antisense RNA, and base/gene editing technologies were gradually developed to improve the quality of rice starch [34]. Zhao (2007) used RNAi technology to interfere with *GBSSⅡ*, *Sbe3*, *SSⅠ*, *SS*Ⅱ, *Wx*, *PUL,* and *ISA* genes in rice, and created breeding intermediate materials with different AC, GC, and GT. The analysis of the ECG of transgenic plants showed that the AC of all mutants was decreased to various degrees, and most of them reached a significant level [35]. Terada (2000) used antisense RNA technology to introduced the antisense *Wx* gene into japonica and indica rice, and obtained the transgenic lines with ACs significantly reduced [36]. Mao (2013) used the CRISPR/Cas9 system to conduct gene editing research on Arabidopsis and rice genes [37]. Today, precise genome modifications with the CRISPR/Cas9 tool have revolutionized genome editing research [38]. It is a more efficient, accurate, easier, and cost-effective technique for achieving gene knock-out in a cell [39] and has made preliminary progress in the improvement of rice quality, especially in the research of *Wx* gene editing.

In this study, the CRISPR/Cas9 system was used to edit the 5′UISS of the *Wx* gene in indica rice lines MMB, TKB, and dr462. The sequencing of the target site in three materials showed consistency with the design target (Figure 1), and all of them performed efficient and accurate gene editing. This seems impossible for traditional breeding and other gene editing technologies. In addition, the mutant lines in the T_2_ generation obtained in this study had decreased AC, increased GC, increased BDV, decreased SBV, and decreased CSV, without significant changes to their main agronomic traits. It is presented that the desirable traits can be stably inherited to the selfed offspring, indicating that CRISPR/Cas9 technology provides an effective strategy for improving the ECQ of rice.

## 4. Materials and Methods

### 4.1. Plant Materials and Growth Conditions

Three elite indica lines were selected, including maintainer Meimeng B (MMB), Tiankang B (TKB), and restorer DR462, which had high levels of AC. The seeds were supplied by the State Key Laboratory of Conservation and Utilization of Subtropical Agri-bioresources, Guangxi University. All rice lines were grown in a mesh room with isolation conditions in Nanning, China, during the normal rice-growing seasons, and treated according to the conventional planting management method.

### 4.2. Construction of CRISPR/Cas9 Vectors and Screening of Homozygous Mutants

The CRISPR/Cas9 targeting vector was constructed for editing the expression vector with reference to a previous study [16]. Vectors and promoters pYLCRISPR/Cas9Pubi-H and pYLsgRNA-LzU6a were provided by the State Key Laboratory of Conservation and Utilization of Subtropical Agri-bioresources, South China Agricultural University. Bacterial strain *E. coli* DH5α for plasmid propagation and preservation and agrobacterium EHA105 for genetic transformation were provided by the State Key Laboratory of Conservation and Utilization of Subtropical Agri-bioresources, Guangxi University.

According to the *Wx* (*LOC_Os06g04200*) genomic sequence provided by the China Rice Data Center [40]. The target sites in the *Wx* gene were designed via the CRISPR-GE [41] online toolkit and cloned into sgRNA, as previously described [16]. The constructs were respectively introduced into MMB, TKB, and DR462 by *Agrobacterium*-mediated transformation. The genomic DNA of transformed plants was extracted from young leaves using the CTAB method [42]. The target sites of the T_0_, T_1_, and T_2_ generations were sequenced using the Sanger method [43]. The PCR products were detected using agarose gel electrophoresis. The homozygous *Wx* mutants without the T-DNA insertion were screened and selected for further experimental analysis. The primers used are listed in Table 5.

### 4.3. Quantitative Real-Time PCR (qRT-PCR) Expression Analysis

Total RNA was extracted from rice caryopses (with glumes removed) 10 days after flowering (DAF) using an RNAplant Plus Reagent kit (Tiangen, Beijing, China). First-strand cDNA was synthesized using a PrimeScript RT reagent kit (Takara, Tokyo, Japan), and qRT-PCR was performed on a CFX Connect Real-Time PCR Detection System (Bio-Rad, Hercules, CA, USA) using Cham Q SYBR qPCR master Mix (Vazyme, Nanjing, China). The Actin gene was used for normalization, and each experiment included three biological replicates.

### 4.4. Measuring Rice Grain Physicochemical Properties

AC, GC, and RVA were measured as described in the national agricultural industry standard (NY/T83-2017) [44]. In brief, AC was determined using a dual-wavelength spectrophotometric method, by drawing the reference wavelength analysis curve for AC and amylopectin (Figure 10A) and the standard curve for amylose (Figure 10B).

GC was determined by measuring the length of the rice gel after gelatinization and cooling. RVA spectrum was measured using a Techmaster RVA instrument (Pertentecmarster, Sweden, Nanning, China). The primary RVA parameters included PKV, BDV, SBV, CSV, HPV, and CPV. All tests were performed in triplicate.

### 4.5. Agronomic Trait Investigation

To detect the agronomic traits of homozygous mutants in the T_2_ generation, together with the corresponding wild type, 20 plants from each line were subjected to agronomic trait measurement at the maturation stage. The traits including plant height, thousand-grain weight, panicle length, grains per panicle, seed setting rate, and chalkiness degree. Field management followed normal agronomic practices.

### 4.6. Statistical Analysis

At least three replicates were performed for each experiment. Statistical and graphical analyses were performed using Excel 2016, GraphPad Prism 9, and Photoshop 7.0. All data were expressed as means  ±  standard deviations (means  ±  SD). One-way analysis of variance (ANOVA) was used to determine the level of significance (* and ** indicate significant differences at *p*  <  0.05 and *p*  <  0.01, respectively). Different lower-case letters indicate statistically significant differences at *p*  <  0.05.

## 5. Conclusions

In this study, the elite indica rice varieties MMB, TKB, and DR462 were chosen as experimental materials, and the 5′UISS in the 5′UTR region of *Wx* gene was edited by CRISPR/cas9 gene editing technology. Screening was carried out to achieve homozygous gene-edited lines, and the relevant phenotypes of these lines were evaluated.

1.In the T_0_ generation of transgenic plants, the genetic transformation efficiency was more than 80%. The sequencing analysis showed that, among the four mutation types, homozygous mutation occupied a relatively large proportion, which not only demonstrated the extremely high editing efficiency of the CRISPR/cas9 gene editing technology, but also implied a great potential for application in designing targets in the 5′UTR region;2.The expression analysis showed that the expression level of *Wx* gene in the edited lines was significantly lower than that of their wild type parents, indicating that editing 5′UISS changed the splicing mode and efficiency of introns;3.The ECQ analysis of MMB-10-2, TKB-21-12, and DR462-9-9 showed that, compared with the wild type parents, the AC of the editing lines was significantly reduced, reaching the first-grade high-quality standard concerning AC. The GC of the edited lines was significantly increased, far exceeding the first-grade high-quality standard concerning GC. The RVA spectrum of the edited lines showed that the grain of the edited lines became softer and more elastic, and the eating quality was greatly improved.4.The main agronomic traits of MMB-10-2, TKB-21-12, and DR462-9-9 had no significant difference from their corresponding wild type, which implied that the trait changes caused by homozygous mutations could be stably inherited to self-bred offspring, reflecting the stability and reliability of CRISPR/cas9 gene editing technology in rice genetic improvement.

## Figures and Tables

**Figure 1 ijms-23-10517-f001:**
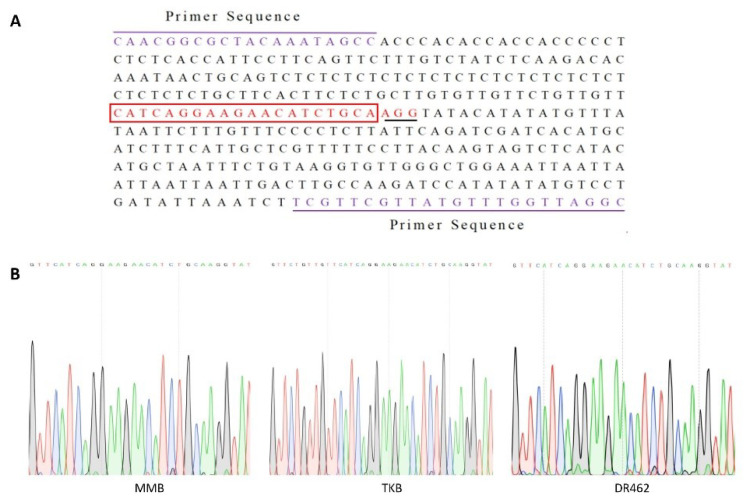
Location of sgRNA in the *Wx* gene and designed primers for amplification. (**A**): *Wx* gene partial sequence showing target positions and primer sequences; purple nucleotides: Upstream and downstream primers; Red Box: sgRNA sequences; black underline nucleotides: protospacer adjacent motif (PAM) region; (**B**): Sequencing peak map of sgRNA.

**Figure 2 ijms-23-10517-f002:**
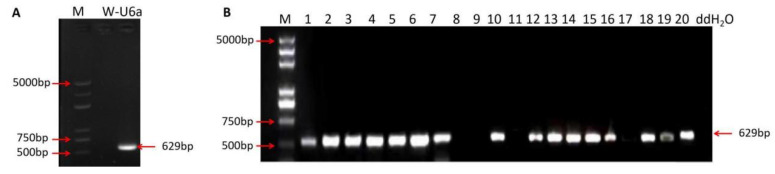
Construction of vector and verification of sgRNA. (**A**): Detection results of sgRNA expression cassette; (**B**) Verification of the size of *Wx* gene cas9/sgRNA expression vector segment; M: DL 5000 DNA marker; W-u6a: sgRNA-u6a fragment; ddH_2_O is a negative control based on sterile water; 1–20 are the numbers of the positive clone bacterial solution picked at random; Lane 8, 9, 11, and 17 are empty plasmids; 5000 bp, 750 bp, 500 bp, 629 bp: fragment size.

**Figure 3 ijms-23-10517-f003:**
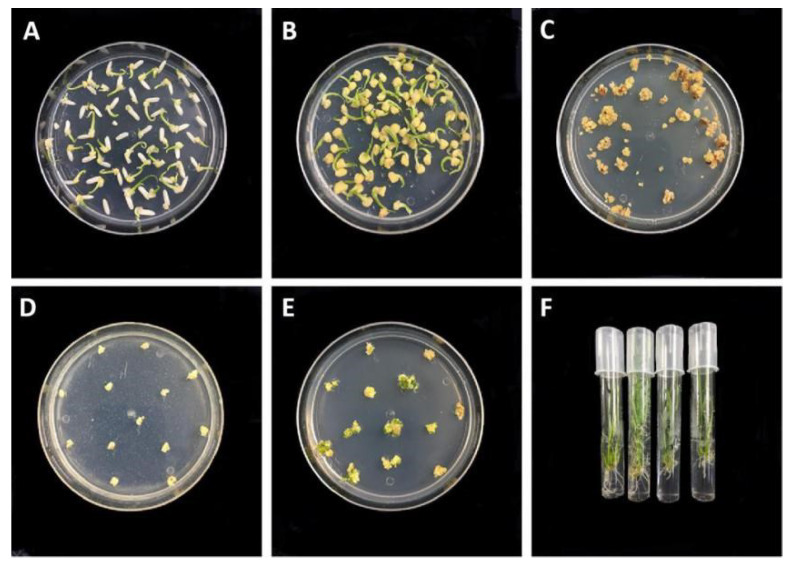
Process steps of genetic transformation. (**A**): Callus induction; (**B**): Callus screening; (**C**): The first screening of calli; (**D**): The second screening of calli; (**E**): Plantlet differentiation; (**F**): Rooting.

**Figure 4 ijms-23-10517-f004:**
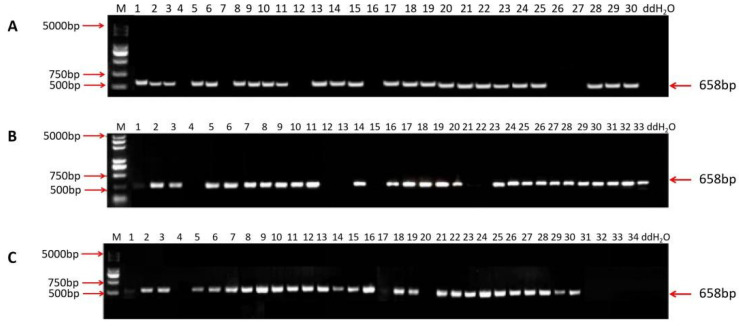
Amplification results of T_0_ generation transformed plants of MMB (**A**), TKB (**B**), and DR462 (**C**). M: DL 5000 DNA Marker; ddH2O: a negative control based on sterile water; 5000 bp, 750 bp, 500 bp, 658 bp: fragment size.

**Figure 5 ijms-23-10517-f005:**
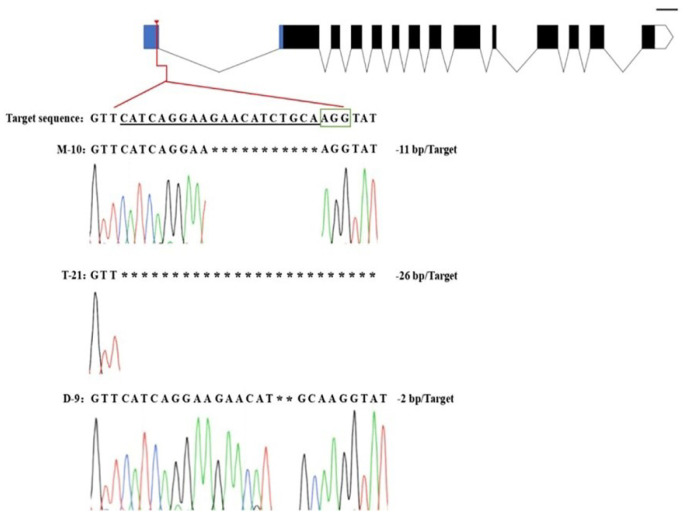
Target mutation types of three homozygous mutants in T_0_ generation. M-10: Mutant MMB-10 from MMB; T-21: TKB-21 from TKB; D-9: DR462-9 from DR462: Green box: PAM: *: missing base.

**Figure 6 ijms-23-10517-f006:**
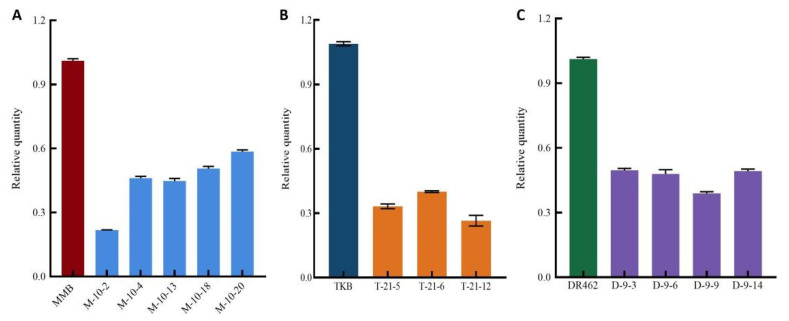
Relative expression level of *Wx* gene in mutant lines in T_2_ generation and their respective wild type. (**A**): MMB; (**B**): TKB; (**C**): DR462; In the mutant lines, M, T, and R are short for MMB, TKB, and DR462.

**Figure 7 ijms-23-10517-f007:**
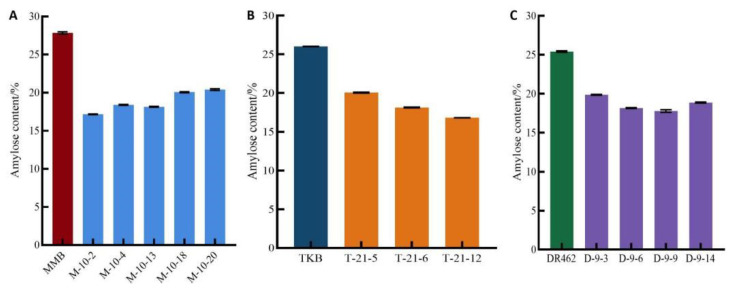
Amylose content in T_2_ homozygous mutant lines and their respective wild types (**A**): MMB; (**B**): TKB; (**C**): DR462; In the mutant lines, M, T, and R are short for MMB, TKB, and DR462.

**Figure 8 ijms-23-10517-f008:**
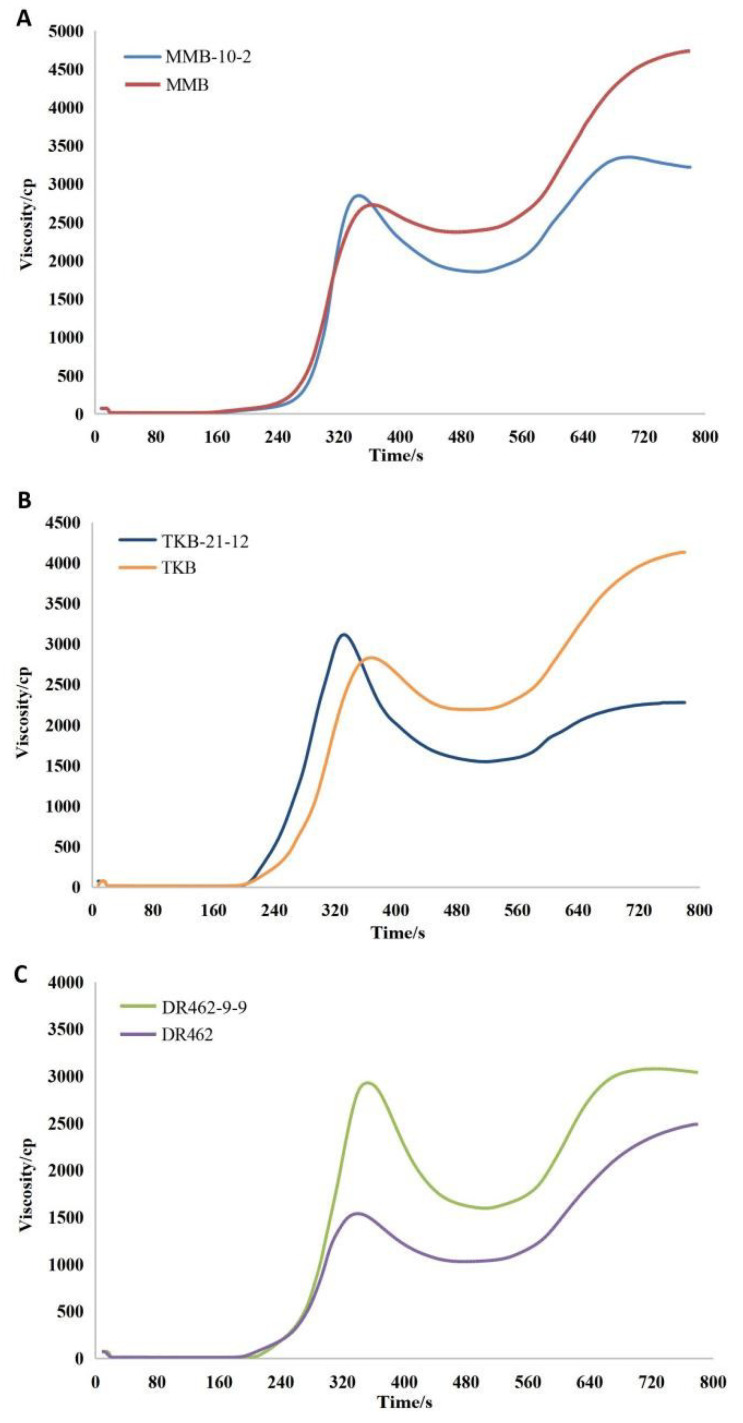
Comparison of RVA profiles between mutant lines in the T_2_ generation and their respective wild types. (**A**): MMB; (**B**): TKB; (**C**): DR462.

**Figure 9 ijms-23-10517-f009:**
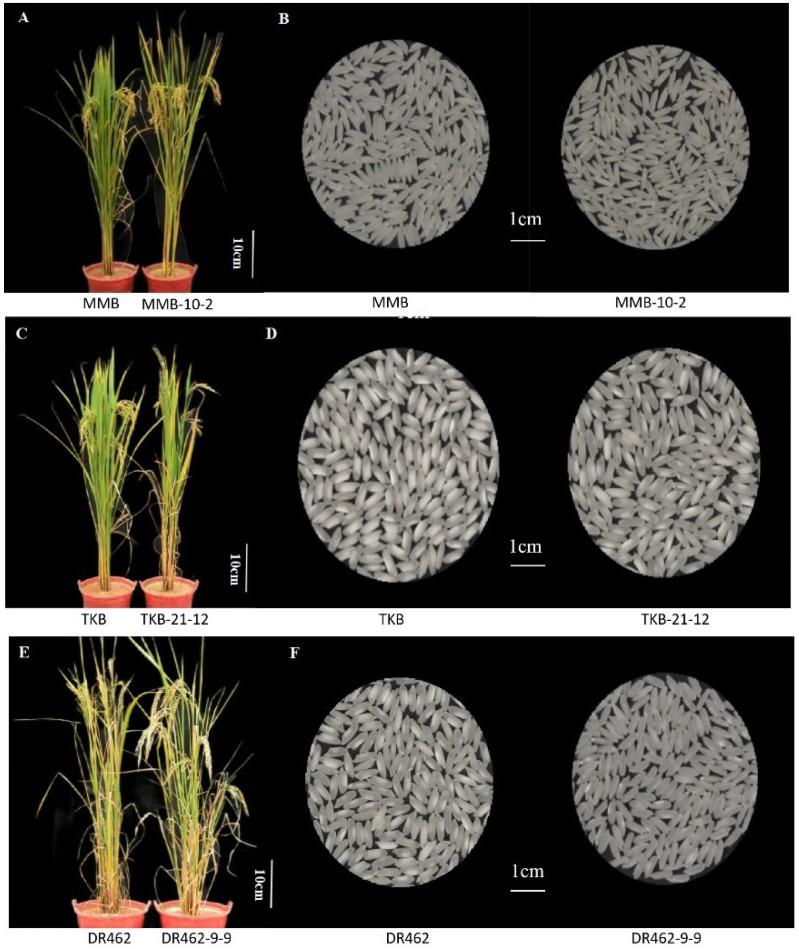
Comparison of plant type and polished rice appearance between mutant lines in the T_2_ generation and their respective wild types. (**A**,**B**): Comparisons between MMB and MMB-10-2; (**C**,**D**): Comparisons between TKB and TKB-21-12; (**E**,**F**): Comparisons between DR462 and DR462-9-9.

**Figure 10 ijms-23-10517-f010:**
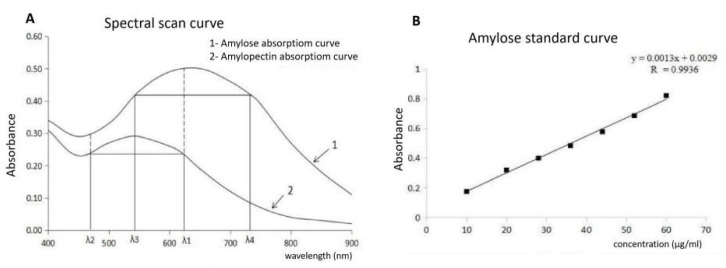
Drawing the standard curve of amylose. (**A**): Preparation analysis of amylose and amylopectin (**B**): Standard curve of amylose.

**Table 1 ijms-23-10517-t001:** Analysis of mutation types in the T_0_ generation.

Parent	Mutation Type									
	Homozygous		Bi-Allelic		Heterozygous		Wild		Total	
	No. of Plants	Rate (%)	No. of Plants	Rate (%)	No. of Plants	Rate (%)	No. of Plants	Rate (%)	No. of Plants	Rate (%)
MMB	8	33.3	7	29.2	4	16.7	5	20.8	24	100
TKB	9	34.6	7	26.9	5	19.2	6	23.1	26	100
DR462	9	34.6	8	30.8	5	19.2	4	15.4	26	100

**Table 2 ijms-23-10517-t002:** GC in partial T_2_ mutant lines and wild type.

Wild Type	GC (mm)	Mutant Lines	GC (mm)
MMB	60.2 ± 0.2	MMB-10-2	78.6 ** ± 0.1
TKB	61.2 ± 0.2	TKB-21-6	77.4 ** ± 0.4
DR462	61.4 ± 0.6	DR462-9-9	79.6 ** ± 0.5

Note: ** *p* < 0.01 derived from one-way ANOVA with LSD test.

**Table 3 ijms-23-10517-t003:** RVA profile characteristics of rice starch mutant lines in the T_2_ generation and their respective wild types.

Lines	PKV	BDV	SBV	CSV	HPV	CPV
MMB	2729.6 ± 43.0	355.0 ± 6.5	2353.6 ± 21.5	2708.6 ± 26.8	2374.7 ± 40.8	5083.3 ± 37.2
MMB-10-2	2850.3 ± 39.0	994.0 ± 9.0	1366.3 ± 11.5	2360.3 ± 29.1	1856.3 ± 24.0	4216.7 ± 50.0
TKB	2832.3 ± 10.1	641.3 ± 22.5	1978.0 ± 14.7	2619.3 ± 36.1	2191.0 ± 28.1	4810.3 ± 39.7
TKB-21-12	3116.0 ± 6.03	1533.6 ± 15.1	729.6 ± 4.5	2263.3 ± 14.4	1539.0 ± 11.7	3802.0 ± 39.0
DR462	1539.6 ± 38.1	508.7 ± 18.0	1457.6 ± 19.8	1966.3 ± 24.5	1031.0 ± 25.0	2997.3 ± 49.2
DR462-9-9	2875.3 ± 34.6	1293.3 ± 21.0	1444.6 ± 3.5	2738.0 ± 21.1	1582.0 ± 14.1	4320.0 ± 35.0

Note: PKV: peak viscosity; BDV: breakdown viscosity; SBV: setback viscosity; CSV: consistency viscosity; HPV: hot paste viscosity; CPV: cool paste viscosity.

**Table 4 ijms-23-10517-t004:** Agronomic traits of the mutant lines in the T_2_ generation.

Lines	Plant Height (cm)	1000-Grain Weight (g)	Panicle Length (cm)	Grain Number per Panicle	Grain Set Rate (%)	Effective Spike Number
MMB	83.9 ± 1.3 a	19.4 ± 3.1 a	27.8 ± 0.1 a	186 ± 0.1 a	87.4 ± 2.0 a	8.5 ± 2.5 a
MMB-10-2	85.1 ± 0.9 a	19.1 ± 2.8 a	27.7 ± 0.3 a	196 ± 0.1 a	87.2 ± 2.5 a	8.7 ± 2.7 a
TKB	77.8 ± 1.2 b	27.4 ± 2.5 b	27.6 ± 0.3 b	190 ± 0.1 b	86.8 ± 1.8 b	8.2 ± 1.8 b
TKB-21-12	76.8 ± 2.1 b	27.0 ± 3.0 b	27.6 ± 0.3 b	186 ± 0.1 b	87.0 ± 2.0 b	8.3 ± 1.4 b
DR462	94.7 ± 1.6 c	24.1 ± 3.1 c	28.0 ± 0.2 c	188 ± 0.1 c	86.6 ± 1.9 c	7.9 ± 2.0 c
DR462-9-9	93.7 ± 1.4 c	23.8 ± 2.9 c	27.9 ± 0.2 c	180 ± 0.1 c	85.9 ± 1.8 c	7.7 ± 2.5 c

Note: a, b, c, *p* = 0.05 level, no significant difference in each line.

**Table 5 ijms-23-10517-t005:** Primer sequences used in this study.

Primer Name	Primer Sequence 5′-3′
*Wx*-text-F	TCCGCCACGGGTTCCAG
*Wx*-text-R	CTCCTACCTCAGCCACAACG
U-F	CTCCGTTTTACCTGTGGAATCG
gR-R	CGGAGGAAAATTCCATCCAC
*Wx*-U6a-1F	TGTGTGCTTACAGCCATGGCGTTTTAGAGCTAGAAAT
*Wx*-U6a-1R	GCCATGGCTGTAAGCACACACGGCAGCCAAGCCAGCA
Pps-GGL	TTCAGAGGTCTCTCTCGACTAGTATGGAATCGGCAGCAAAGG
Pgs-GGR	AGCGTGGGTCTCGACCGACGCGTATCCATCCACTCCAAGCTC
PB-R	GCGCGCGGTCTCTACCGACGCGTATCC
PB-L	GCGCGCgGTCTCGCTCGACTAGTATGG
HPT-F	ATTTGTGTACGCCCGACAGT
HPT-R	GTGCTTGACATTGGGGAGTT
CAS9-F	CTGACGCTAACCTCGACAAG
CAS9-R	CCGATCTAGTAACATAGATGACACC

Note: ACTAGT and ACGCGT are *Spe*Ⅰ and *Mlu*Ⅰ sites, respectively.

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
