# Peer review of "Development of Soft Rice Lines by Regulating Amylose Content via Editing the 5′UTR of the Wx Gene"

_ijms, 2022, doi:10.3390/ijms231810517_

Round 1

Reviewer 1 Report

The article is interesting and innovative. The results can be useful for the food industry worldwide. Please check spelling and English language errors.

Author Response

Dear reviewer:

Thank you for your decision and constructive comments on our manuscript. Corresponding revisions have been made according to the comments:

1-12: The formatting and spelling errors pointed out in the comment have been revised.

13:  The English language was rechecked.

Sincerely,

Jinlian Yang

Reviewer 2 Report

The current manuscript reports the development of soft rice lines by regulating amylose content via editing the 5'UTR of the Wx gene.

In general, this is an important and interesting work. I have, however a few comments or suggestions.

Line 34: A dot is missing at the end of the sentence.

In the last paragraph of the Introduction, I recommend formulating a specific goal of the research.

The structure of the scientific article is broken, the second section should describe the materials and methods, and therefore it is very difficult to immediately perceive the results of the research. The results are not perceived at all, because all abbreviations used are listed below in the Materials and Methods section

Line 304: bacterial genus not in italics

Not all types of equipment used have a brand and manufacturer

There are no conclusions.

Author Response

Dear reviewer:

Thank you for your decision and constructive comments on our manuscript. Corresponding revisions have been made according to the comments:

1: The formatting errors in Line 34 have been revised.

2: As suggested, a specific goal of the research was revised in the last paragraph of the Introduction.

3: The structure of this article follows the guidelines of the journal, which requires that the results be placed in the second section. And we rechecked where the abbreviation was first used, to make sure it accompanied by the full name.

4: At Line 304, the “bacterial genus” referred to in the comment was not found.

5: The brand and manufacturer of each equipment were rechecked.

6: Conclusion has been supplemented.

Sincerely,

Jinlian Yang
